# Model-Free Approach for the Configurational Analysis of Marine Natural Products [note 1]

**DOI:** 10.3390/md19060283

**Published:** 2021-05-21

**Authors:** Matthias Köck, Michael Reggelin, Stefan Immel

**Affiliations:** 1Alfred Wegener Institute, Helmholtz Centre for Polar and Marine Research, Am Handelshafen 12, 27570 Bremerhaven, Germany; 2Clemens-Schöpf-Institute for Organic Chemistry and Biochemistry, Technical University of Darmstadt, Alarich-Weiss-Straße 4, 64287 Darmstadt, Germany; re@chemie.tu-darmstadt.de

**Keywords:** configurational analysis, distance geometry, distance bounds driven dynamics, NMR spectroscopy, NOE data

## Abstract

The NMR-based configurational analysis of complex marine natural products is still not a routine task. Different NMR parameters are used for the assignment of the relative configuration: NOE/ROE, homo- and heteronuclear *J* couplings as well as anisotropic parameters. The combined distance geometry (DG) and distance bounds driven dynamics (DDD) method allows a model-free approach for the determination of the relative configuration that is invariant to the choice of an initial starting structure and does not rely on comparisons with (DFT) calculated structures. Here, we will discuss the configurational analysis of five complex marine natural products or synthetic derivatives thereof: the *cis*-palau’amine derivatives **1a** and **1b**, tetrabromostyloguanidine (**1c**), plakilactone H (**2**), and manzamine A (**3**). The certainty of configurational assignments is evaluated in view of the accuracy of the NOE/ROE data available. These case studies will show the prospective breadth of application of the DG/DDD method.

## 1. Introduction

The determination of the relative configuration of marine natural products usually follows the constitutional structure elucidation. Despite numerous attempts, there is so far no standard or even automated protocol for the configurational and conformational assignment of natural products or small organic molecules in general [1,2,3,4,5,6,7,8,9,10,11,12,13,14]. Although recently, the application of anisotropic NMR data, such as residual dipolar couplings (RDCs) or residual chemical shift anisotropy (RCSA), has become more and more popular, the cross-relaxation (NOE or ROE) from which interproton distances can be obtained is still a major source of structural information. The use of NOE- or ROE-derived interproton distances for the configurational analysis is a mature technique [15,16] that can be amended by including RDCs in restrained distance geometry (rDG) simulations in a protocol that we have discussed in detail very recently [17]. Herein, we will concentrate on the NOE/ROE-restrained distance geometry [18,19,20] and distance bounds driven dynamics method (rDG/DDD) [21,22], which is, from our point of view, the method of choice for the configurational analysis of small organic molecules with several stereogenic centers [23,24,25,26,27,28]. The use of interproton distances from NOESY spectra as restraints in molecular dynamics simulations (rMD) was introduced about 35 years ago [29,30] and since then, it has been a successful method for the NMR-based conformational analysis of biomacromolecules. The DDD approach was later applied as a method for “shaking up” the structures before starting the rMD analysis [21,22]. The concept of floating chirality (fc) was added to aid the assignment of diastereotopic protons or methyl groups in proteins, and was first applied in 1988 to distance geometry (DG) calculations [31] and in 1989 to rMD simulations [32]. The most important aspect of the rDG/DDD method is the possibility to allow configurations to dynamically change during the simulation (floating chirality, fc) and therefore to determine the conformation and the relative configuration of small organic molecules, simultaneously (fc-rDG/DDD).

This approach combines distance geometry (DG) [18,19,20] and distance bounds driven dynamics (DDD) [21,22], using NOE/ROE-derived distances as restraints [23,24,25,26,27,28]. The general workflow of the DG/DDD method is represented in Figure 1. Similar to using a molecular model construction kit, molecular geometries are generated within the limits of holonomic distance bounds (relations between positional coordinates), derived from the constitution and experimental distance margins deduced from NMR data. As this step a uses distances only, and all implied information on angles and torsions is actually discarded, it is ensured that the structures generated are not biased by the input structure; models of all possible configurations and conformations are generated with purely statistical weights. In order to add additional degrees of freedom to this building process (“metrization”), it is carried out in four-dimensional (4D) Cartesian space. These initial guess models (Figure 1) are then refined by a sequence of 4D and 3D simulated annealing simulations (DDD) where at each step, the squared violations of experimental restraints are computed as a dimensionless pseudo energy (total error) with empirically chosen “force constants”. In all dynamics and optimization calculations, the partial derivatives $\partial E_{\mathrm{total}}/\partial r_{\alpha }$ (α∈x,y,z(,w) for all atoms) of this pseudo energy term, with respect to 4D and 3D Cartesian atomic coordinates, are interpreted and used as forces governing the evolution of the system; a detailed description of this procedure is given in Refs. [17,21,22]. Finally, all emergent structures are ranked and sorted by their pseudo energies (total error; sum of total violations), and the lower this “energy” becomes, the more likely the corresponding molecular geometry represents the correct configuration (and conformation) of the compound under investigation (Figure 1). Note that at any point of any rDG/DDD simulation, neither a conventional parametrized force-field is involved, nor are any presumptions on conformational preferences implied, but all molecular structures and relative configurations of stereogenic centers evolve solely on the basis of experimental NMR data (Figure 1, Steps b and c).

Although DDD calculations do not apply any angle- or torsion-dependent pseudo energy terms, the geometry of tetrahedral or planar centers can be maintained by so-called chiral volume restraints (Figure 2). These are defined as pseudo scalar triple products $V_{\mathrm{chir}}=\vec{a}\cdot \left(\vec{b}\times \vec{c}\right)$ of three vectors spanned by the substituents on each center, respectively. Figure 2 illustrates the definition of chiral volumes in DG/DDD for sp^3^- and for sp^2^-type atomic centers, where the absolute configuration of stereogenic centers is encoded by the sign of $V_{\mathrm{chir}}$, and $V_{\mathrm{chir}}=0$ for planar fragments. Though these parameters can be used in principle as restraints to fix the configuration of certain stereogenic centers in the course of DDD simulations, floating chirality is achieved by not using chiral volume restraints for unknown prochiral and stereogenic centers. In general, we apply chiral volume restraints only to methyl groups to keep them tetrahedral, and to aryl rings and double bonds to bound them to planarity, respectively; only one stereogenic center of choice is fixed in an arbitrary selected absolute configuration in order to avoid indistinguishable enantiomeric structures.

In contrast to rMD simulations, DG relies solely on experimental data, and all stereogenic centers are allowed to adopt their relative configuration in accordance with the experimental data and thus, remove any intrinsic bias imposed on the results by not implementing physical force-fields of any type. As already mentioned, another important aspect of the DG/DDD approach is that all structure models are first generated in four-dimensional (4D) space before these are transferred into “real” 3D space (Figure 1, Step b). The extra dimension not only provides additional degrees of freedom to assemble structures of different configurations and conformations within the limits of the distance bounds, but also efficiently eliminates “energy” barriers for configurational inversions in the DDD step [33,34]. By analogy with 2D chiral objects that can be transformed into each other through a rotation in higher dimensions (3D), the configuration of 3D chiral objects can be inverted by rotations in 4D (Figure 3).

However, there are fundamental differences in treating large biomacromolecules, such as proteins or small (marine) natural products, using the rDG/DDD approach. For the former, usually very large datasets of intramolecular distances can be obtained, and the “configurational problem” can generally be ignored, as the configurations of the repeating units (e.g., amino acids for proteins) are known. Thus, only qualitative distances defined by general categories (e.g., “weak”, “medium”, and “strong”) are sufficient to obtain reasonable 3D models for biomacromolecules. Quite another matter are small molecules with multiple stereogenic centers, where the relative configuration of the constituent units is unknown a priori and only a limited number of NOE contacts can be identified. Therefore, more accurate intramolecular distances and tighter upper and lower distance limits are required to unequivocally deduce the relative configuration of complex natural products.

Generally, the interproton distances are derived by volume integration of the cross-peaks in the 2D NOESY or 2D ROESY spectrum [15,16]. The conversion of volume integrals into distances is carried out using reference peaks related to known intramolecular distances (“fixed distances”). Typical examples are diastereotopic methylene groups or vicinal CH–CH contacts in aromatic systems. All other distances are then derived from the r^−6^-weighted ratios of the corresponding peak volumes with respect to the volume of the reference cross peak. An important point when dealing with NOESY and ROESY spectra is the mixing time of the experiment. The dependence of the NOE or ROE effect with respect to the mixing time is of an exponential nature, which does require measuring build-up rates with several different mixing times. This can be circumvented by using the linear approximation approach (initial rate approximation), which would require measurement with relative short mixing times (linear region of the exponential function). This assumption is valid when the distances extracted for two different mixing times are identical.

Especially for small molecules in the extreme narrowing limit, the accumulation of quantitative NOESY/ROESY spectra also requires carefully degassing the NMR samples by freeze–pump–thaw techniques in order to remove dissolved paramagnetic oxygen. These experimental intricacies and the mixing-time dependency of the effect frequently lead to using only qualitative NOE intensities (“weak”, “medium”, and “strong”) and rather large empirical error estimates on internuclear distances for the configurational analysis of natural compounds. The choice for a NOESY or a ROESY spectrum depends mainly on the magnetic field strength, the size of the molecule, and the solvent viscosity. NOESY spectra have the disadvantage that for medium-sized molecules, the NOE becomes 0 if $\left(\omega_{0}\cdot \tau_{C}\right)\approx 1.12$. This is approximately the case for a correlation time $\tau_{c}$ of 0.3 ns at 400 MHz. ROESY spectra do not have this disadvantage, but because of the necessary spinlock to achieve transverse cross relaxation, other effects (e.g., offset dependence or TOCSY artifacts) have to be considered. In general, ROESY spectra are plagued by more artifacts than NOESY spectra [16]. For the conversion of the NOE/ROE-derived distances into distance restraints (upper/lower bounds), we generally use ±10% (+pseudo-atom [35] correction for positions with more than one isochronous H) in order to account for experimental errors. For compounds **1a**–**c,** we also tested ±5%, ±20%, and ±30% error margins on distances in order to evaluate the certainty of configurational assignments with respect to the accuracy of the experimental data (more details are given in Chapter 2).

In the literature, it is a common standard to assume an uncertainty of ±10% in NOE/ROE-derived distances due to experimental errors (without the consideration of the pseudo-atom correction), though it must be noted that the error-prone quantities measured are actually the volume integrals of the cross peaks. Due to the $r^{-6}$ scaling of the cross-peak volumes, the uncertainties in both quantities peak volumes and distances scale differently, and—given a proper setup of the NMR experiment and correct evaluation of the build-up rates and mixing times—an assumed error of ±10% of the former leads to a drop in the accuracy of the distance information to well below ±5%.

## 2. Results and Discussion

Here, the application of the fc-rDG/DDD method as implemented in ConArch^+^ [36,37,38] will be demonstrated on five complex natural products (Scheme 1). Compounds **1a** to **1c** are synthetic derivatives or congeners of palau’amine, which is probably the most prominent cyclic dimeric pyrrole-imidazole alkaloid (PIA). Palau’amine, a PIA with a hexacyclic core, was originally described in 1993 from the marine sponge *Stylotella agminata* (in 1998 changed to *Stylotella aurantium*) from the Western Caroline Islands (**1d**) [39]. Moreover, the successful application of the method will be shown for two other marine natural products, using data from the literature. The oxygenated polyketide plakilactone H (**2**) was isolated from the marine sponge *Plakinastrella mamillaris* from Fiji in 2013 [40]. The β-carboline alkaloid manzamine A (**3**) was isolated from the marine sponge *Haliclona* spp. from Okinawa, Japan in 1986 [41].

As NMR can determine relative configurations only, in all rDG simulations, a single stereogenic center of compounds **1** to **3** each was fixed by applying a chiral volume restraint in order to avoid enantiomeric structures. In the sequel, the number of the generated structures in the fc-rDG/DDD calculations was set to 1000 to allow for reasonable sampling of the configurational and conformational space.

### 2.1. Palau’amine Derivatives *(**1**)*

Palau’amine itself was a very prominent target in synthetic organic chemistry from 1995 to 2010 [42]. The original structure of palau’amine (**1d**), proposed in 1993 [39], was first revised in 1998 [43], correcting the configuration of C-20 (**1e**). As discovered in 2007, up to that time, all groups followed the wrong target with *cis*-fused rings D and E [44]. In that year, the relative configuration of C-12 and C-17 of the palau’amine congeners was revised and the unusual *trans*-fusion of the five-membered rings was established [45,46,47]. The total synthesis of *rac*-**1f** (Scheme 2) was completed by the Baran group in 2010 [48,49], and the asymmetric variant in 2011 [50].

Compounds **1a** and **1b** are synthetic intermediates [51] on the route to the “old” relative configuration of palau’amine (**1d** and **1e**). From our point of view, these two synthetic derivatives and compound **1c** are perfect candidates for which we have illustrated the “starting structure problem”. The *cis*- and *trans*-configured palau’amine congeners are very well-known examples from the literature (as mentioned before) and therefore, an ideal test case for the evaluation of the fc-rDG/DDD method. Additionally, **1a** and **1b** differ from palau’amine in that there is no chlorine at C-17, which reduces the number of contiguous stereogenic centers from eight to seven, resulting in 64 possible relative configurations (diastereomers), respectively. Furthermore, the short side chain of palau’amine at C-18 (–CH_2_–NH_2_ in **1d**–**f**) is changed to –OH (**1a** and **1b**). Compounds **1a** and **1b** are C-20 epimers, which are the stereogenic centers whose configuration was revised from 1993 (**1d**) to 1998 (**1e**). Tetrabromostyloguanidine (**1c**) has eight contiguous stereogenic centers, resulting in 128 possible relative configurations (diastereomers).

For our investigation on compounds **1a** to **1c**, we used the ROE-derived distances from Overman’s work in 2007 (from the ROESY spectra with a mixing time of 100 ms; **1a**: 16 ROEs, **1b**: 17 ROEs, and **1c**: 9 ROEs) [51]. The experimental setup for **1a** and **1b** was identical to our own setup for tetrabromostyloguanidine (**1c**) in 2007 [45], which is a perfect precondition for a direct comparison of the results. The complete lists of ROEs of **1a**, **1b**, and **1c** are given in the Appendix A. As mentioned above, one stereogenic center of each compound was fixed and set as a reference (C-10 for all three compounds). In a traditional approach of pre-calculating structures, this would entail the necessity to evaluate a total of 64 or 128 diastereomers, respectively. In the fc-rDG/DDD-calculations described here this is not necessary because these configurations evolve under the influence of the distance restraints without any action of the experimentalist. The ROE interproton distances for compounds **1a** to **1c** were used with varying lower and upper distance bounds (±5%, ±10%, ±20%, and ±30%) in order to demonstrate the importance of accurate distance limits for the configurational analysis of small molecules in floating-chirality-restrained DG/DDD calculations (fc-rDG/DDD).

The total error (dimensionless) of the 500 lowest pseudo energy structures for **1a** is plotted for each structure, ordered according to ascending total errors. The first wrong configuration of **1a** (for the standard DG/DDD calculations using ROEs ±10%) with respect to the seven stereogenic centers is No. 93 (green square in Figure 4a). This structure differs from structures #1 to #92 by a change in all configurations, besides C-6. These two relative configurations have a ΔE value of 0.40. The result of this calculation (±10%) is not dependent on the starting structure. Irrespective of the *cis*- or *trans*-configuration as input coordinates for the calculation, the ROE-derived distances of **1a** lead to a *cis*-fused final structure. Softening up the ROE distance bounds to ±20% and ±30% dramatically downgrades the certainty with which the correct configuration of **1a** is identified by the DG/DDD simulations.

The first wrong configuration shifts to structure No. 21 (blue triangle in Figure 4a) or No. 3 (black reversed triangle in Figure 4a), and the ΔE values corresponding to the washed-out energy steps become very small. As expected, these findings clearly indicate that inaccurate datasets are inappropriate to discriminate between possible diastereomeric structures, which is particularly obvious for the ±30% dataset. On the contrary, decreasing the restraining ROE distance intervals to ±5% significantly reduces the number of eligible diastereomeric structures to satisfy the experimental data, and the first wrong configuration identified shifts to structure No. 138 (orange circle in Figure 4a) and simultaneously increases the pseudo energy error penalty to ΔE=0.92. The inset plot in Figure 4a shows the first “energy” steps in more detail (“best 250”) for a better overview of the results. The plot in Figure 4b (top) shows the structures #1 to #92 with the same relative configuration of all stereogenic centers (±10% calculation) and the DG best-fit structure (lowest pseudo energy, bottom). The above example clearly shows that accurate distance information is of the utmost importance to get the correct molecular configuration of the compound under investigation.

The results of the fc-rDG/DD calculations for the synthetic palau’amine derivative **1b** are represented in Figure 4c,d by plots equivalent to those given above for **1a**. Although compounds **1a** and **1b** only differ in the configuration of a single stereogenic center (C-20) and in one ROE (16 versus 17 experimental restraints), the differentiability of the diastereomers of **1b** is much more pronounced, compared to **1a**. The first different configuration of **1b** (for the standard DG/DDD calculations using ROEs ±10%) with respect to the seven stereogenic centers is No. 174 (green square in Figure 4c). This structure differs from structures #1 to #173 by the configuration of C-20. As already observed for **1a**, the certainty with which the correct configuration of **1b** can be identified becomes significantly attenuated the more broadly the restraining boundaries are chosen (Figure 4c). On the contrary, setting tighter distance limits (±5%) to the experimental restraining dataset again improves the reliability of the configurational assignment dramatically, and the first wrong configuration is observed for structure No. 290 (orange circle in Figure 4c) whilst the ΔE value increases from 1.69 to 2.05.

The simulation results for tetrabromostyloguanidine (**1c**) are depicted in Figure 4e in the same way as described above. The first wrong configuration of **1c** (for the standard DG/DDD calculations using ROEs ±10%) with respect to the seven stereogenic centers is No. 175 (green square in Figure 4e), differing from the preceding structures by the configuration of C-16. Similar to the results obtained above for **1a** and **1b**, tighter NOE restraints lead to an increase in the differentiability of the correct configuration of **1c**, and looser restraints lead to a significant loss of information.

It must be stressed that all rDG simulations yielding quickly and with high reliability the correct configurations of the palau’amine derivatives (**1a**–**c**) are actually single, fully automated calculations (one for each compound) and not 64 or 128 individual calculations on alternate diastereomers. Due to the fact that at no point of the simulations a physical force-field is involved, the method finds *cis*- (**1a** and **1b**) or *trans*-fused (**1c**) five-membered rings (bicyclo [3.3.0] octane ring systems, Scheme 1 and Scheme 2, and orange colored structure fragments in the plots of Figure 4b,d,f) without bias toward the energetically strong-favored *cis*-fusion. Unlike simulations employing physical force-fields, which inevitably would tend to favor the low-energy *cis*-connectivity of the fused rings, the DG simulations do not suffer from this energetic bias, but only follow the restraints imposed by the experimental data. In particular, the initial 4D “metrization” step that antedates any rDG/DDD simulation ensures that the DG results do not depend on an assumed initial configuration, as any biased information in this respect is actually discarded at the earliest possible point of each simulation.

### 2.2. Plakilactone H *(**2**)*

The polyketide plakilactone H (**2**), which was isolated from the marine sponge *Plakinastrella mamillaris,* was chosen as the next example. We used the distance information from the original publication on the structural elucidation of **2** (25 interproton distances extracted from 1D NOESY spectra with mixing times of 500 ms; the complete list of NOEs of **2** is given in the Appendix A) [40].

Using the same methodology as in case of the palau’amine derivatives (**1**), the results for the 1000 generated possible structures of plakilactone H (**2**) are shown in Figure 5a (“best 400”) as a graphical representation of the total error (dimensionless) for each structure, ordered according to ascending total errors; the C-4 quaternary stereogenic center of **2** was set as reference and fixed by the application of a chiral volume restraint.

In the low-pseudo energy region of structures identified for **2** (“best 100” inset plot of Figure 5a) six small energy steps are present, which in total correspond to two configurational families A (#1 to #42, in total 42 structures) and B (#43 to #72, in total 30 structures) separated by a very small energy difference with ΔE=0.10. Both families are subclassified by even smaller energy jumps due to different rotamers of the ethyl groups, which arise from the impossibility to assign the diastereotopic methylene protons a priori and individually in this example. All structures of groups A and B feature the same relative configuration of the stereogenic centers C-6, C-7, and C-8, but differ in their configuration of C-4 only (see structures shown in Figure 5b). Noteworthy, as a side note, is the fact that the structures in group B actually emerged as mirror images, because we fixed the configuration of C-4 in the calculations presented here, and were inverted for plotting in Figure 5b for clarity only. In Figure 5a, additional ethyl rotamers and C-10 epimers intermixed with each other are found along the first larger step of pseudo energy ranked structures (#73 to #280), followed by another step that is associated with the first wrong side chain configuration (structure No. #281, bold green circle in Figure 5a).

In summary, the DG/DDD method unequivocally identifies the correct side chain configuration of **2** (stereogenic centers C-6, C-7, and C-8) on the basis of the NOE data. As displayed by the structures plotted in Figure 6, this approach simultaneously also detects ambiguities in the configurational assignment of C-4 (marked in orange). Indeed, due to the flexibility of the ethyl groups, the experimental data are commensurable with either the 4*R* (epimer A) or 4*S* configuration (epimer B) with about the same error margins.

### 2.3. Manzamine A *(**3**)*

The marine natural product manzamine A (**3**) was chosen as a very complex structure with a relative small number of stereogenic elements (five stereogenic centers plus two double bonds) [41]. For manzamine A (**3**), we used a hybrid dataset based on only nine experimental NOE contacts listed in [41]. Since the authors only have given the interactions of the protons and not the actual distances between them, we used these nine interactions and added the distances from the crystal structure of **3** [41] to obtain the distance restraints for the DG/DDD calculations (the complete list of NOEs of **3** is given in the Appendix A).

The results for the 500 low-energy structures out of 1000 generated structures of manzamine A (**3**) are shown in Figure 7a (in the same representation as for **1** and **2**; C-12 fixed and set as reference). However, the limited NOE dataset and the pronounced conformational flexibility of the 8- and 13-membered rings in **3** make an unambiguous configurational assignment unfeasible, as the pseudo energy steps and differences between the best-fit (correct) and first alternate (wrong, No. #267) configuration vanish (Figure 7a, black line, ΔE=0.10). This means that even with this sparse dataset, the correct configuration was found in the lowest energy structures but due to the missing energy jumps between these structures and the wrong ones, no reliable decision could be made.

For demonstration purposes, we have, therefore, decided to add artificial residual dipolar couplings (RDCs) to the dataset of **3**. The RDCs were calculated from the crystal structure of **3** [41] in conjunction with a randomly generated alignment tensor [36] in order to obtain a total of 28 RDCs (the complete list of RDCs of **3** is given in the Appendix A; the dataset included all methine and methylene protons, where the latter were used only as the corresponding sums of two C-H dipolar couplings without individual diastereotopic assignments). The green line in Figure 7a displays the result for the combined NOE and RDC dataset. Here, the first wrong configuration of **3** is structure No. 422 (green circle in Figure 7a), which differs from structures #1 to #421 by the relative configuration of C-12. By analyzing the structures with correct relative configuration, two conformational families A (185 structures) and B (142 structures) of **3** can easily be identified (Figure 7a). The plot in Figure 7b (top, left) shows the superimposed structures #1 to #185 (conformational family A) of manzamine A (**3**) and also the structures #186 to #327 (top, right; conformational family B). All structures with the correct configuration of **3** (420 structures) are shown in Figure 7b (bottom).

The results obtained by the RDCs-only calculation are shown in the inset plot of Figure 7a (“best 100”, blue line). The first wrong configuration of **3** is structure No. 58 (blue triangle in Figure 7a, inset plot). This structure differs from structures #1 to #57 by the configuration of C-34. This result, together with the one of the NOE-only calculation demonstrates the usefulness of the combined use of NOEs and RDCs for the configurational analysis of manzamine A (**3**). It is obvious that this holds for most other configurational assignment problems.

## 3. Methods

### 3.1. NMR Data

The ROE-derived distance restraints for compounds **1a**–**c** were taken from Ref. [51]. The interproton distances were used from the ROESY spectra with a mixing time of 100 ms. For all compounds, three different mixing times (100, 150, and 200 ms) were measured. The total numbers of ROEs are 16 for **1a**, 17 for **1b**, and 9 for **1c**. The complete lists of ROEs for compounds **1a**–**c** are given in the Appendix A. In the case of **2** only, 1D NOESY spectra with a mixing time of 500 ms were recorded, which led to 25 NOEs [40]. The complete list of NOEs of **2** is given in the Appendix A. The base of the used NOEs for **3** was Figure 2 (“selected NOE correlations”) from Ref. [41]. Since the authors only have given the interactions of the protons and not the actual distances between them, we used these nine interactions and added the distances from the crystal structure of **3** [41] to obtain the distance restraints. The complete list of NOEs of **3** is given in the Appendix A.

### 3.2. DG/DDD

The distance geometry calculations were carried out, using a modified version of the DG-II program (kindly provided by Prof. Ruud Scheek, University of Groningen, the Netherlands), linked and interfaced to our ConArch^+^ a program package. ConArch^+^ is developed by us for the determination of configurations, using arbitrary combinations of all kinds of NMR data (isotropic and anisotropic NMR parameters) as input [36]. The lists of distance restraints for the DG calculations were constructed from the initial guess primary structures, using bond-lengths-derived upper and lower distance bounds (taken as ±1%), respectively; a detailed description of the DG methodology is given in Ref. [17]. The experimental interproton distances derived from NOESY or ROESY spectra were utilized with default error margins of ±10%, respectively. For each DG simulation, 1000 structures were embedded in 4D space used to initiate the distance bounds driven dynamics (DDD) calculation, beginning at 300 K in 4D for 5 ps (5000 steps, step size 10 fs) and then with a gradual reduction in temperature to 0 K over the next 5 ps (5000 steps, step size 10 fs). This complete procedure was repeated in 3D after reduction of the dimension (4D to 3D). More details on the setup of DG/DDD calculation are given in Refs. [17,36].

## 4. Conclusions

As demonstrated in the first part of this work, a reliable determination of relative configurations depends strongly on careful quantification of the NOE/ROE data. The frequently used coarse classification of NOEs/ROEs [54] as being “strong (2.00–2.49 Å)”, “medium (2.50–2.99 Å)”, or “weak (3.00–4.00 Å)” does not fulfil this demand. This classification corresponds to the large NOE-derived distance ranges (±20% and ±30%) we used with the palau’amine derivatives (**1**). The results clearly show that the differentiability of diastereomers dramatically drops if the distance ranges exceed ±10% considerably. Another problem related with the coarse classification of distances is that very similar distances (e.g., 2.45 Å and 2.55 Å) could be assigned to different ranges (Figure 8), which will adversely affect the search for the correct configuration.

The application of the fc-rDG/DDD method to three structural types of marine natural products gave insights into the prospective breadth of this approach. Contrasting methods relying on physical force-fields calculations on the palau’amine derivatives (**1**) have proven that fc-rDG/DDD can identify high-energy *trans*-fused five-membered rings when the data demands this configuration. The relative configurations of all three palau’amine derivatives were unambiguously determined, irrespective of the starting structure with the method discussed. In particular, the results for tetrabromostyloguanidine (**1c**) are truly astonishing since it has eight stereogenic centers and only nine ROEs were sufficient for the correct configurational analysis. Comparing the size and complexity of the two molecules (**1c** and **2**) as well as the number of experimental restraints, one would probably argue that the results for plakilactone H (**2**) should even be better than those for tetrabromostyloguanidine (**1c**). In spite of the comparable large number of NOEs (25) in relation to the size of the molecule, the NOEs were not able to unambiguously solve the relative configuration of **2**. Three of the four stereogenic centers could be determined but it was impossible to assign the quaternary stereogenic center C-4, which means that the analysis proposes two diastereomers (C-4 epimers) as equivalent solutions. The main difference between **1c** and **2** is the conformational flexibility. The obtained results are in accordance with the expectation that flexibility requires more distance restraints. Opposite to **2**, manzamine A (**3**) is a very complex structure for which only a rather limited number of NOE-derived distances (9) were available. When using the same approach as that for **1** and **2**, the configuration of **3** cannot be solved by NOEs only. Only the consideration of RDCs in combination with the NOEs allows for an unambiguous assignment of the five stereogenic centers and the two double bonds.

NMR parameters as NOEs/ROEs and RDCs depend on the measurement conditions (temperature, type of a solvent, concentration, and pH), and the results of the rDG simulations are valid for the conditions under which the NMR data have been recorded. Nevertheless, with changing NMR restraints, the rDG approach can be used to track conformational changes of the analyte.

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
