# Peer review of "Model-Free Approach for the Configurational Analysis of Marine Natural Products†"

_marinedrugs, 2021, doi:10.3390/md19060283_

Round 1

Reviewer 1 Report

Determination of the configuration of the stereocenters of natural compounds is still regarded as a special task, particularly in cases of small molecules contained unusual carbon skeleton. In the manuscript combined distance geometry (DG) and distance bounds driven dynamics (DDD) method is regarded as an approach for the structural analysis of several natural molecules derived from marine animals, as well as of related synthetic compounds. The values of NOE and ROE together with the values of the derived distances of the studied compounds were determined previously and described in a literature. The authors used these data for the development of the theoretical method to predict the configuration of the stereocenters in the studied compounds. Scopes and limitation of the method were demonstrated on the examples: the cis-palau’amine derivatives 1a and 1b, tetrabromostyloguanidine (1c), plakilactone H (2), and manzamine A (3). The results obtained are of interest of scientists deal with structural analysis of natural compounds and could be published in Marine Drugs.

Question:

The values of the NMR data (chemical shifts, NOE, ROE, constants) depend on such parameters as temperature, type of a solvent, concentration, pH. Is it possible to take into account these parameters in the calculations?

Author Response

An extra paragraph at the end of conclusion was added to consider the question of reviewer 1.

Reviewer 2 Report

The fc-rDG/DDD method applied to the elucidation of relative stereochemistry of marine natural products is described in this manuscript. Although the complete elucidation of relative stereochemistry was not obtained in the cases of plakilactone H and manzamine, the methodology was proven to work well with palau’amine derivatives. A part of the manuscript-which describes the concept and principle of the method- shows overlap with 2020 Marine Drugs paper by the same authors, however, the results described in the manuscript clearly showed the expansion in the scope of the methodology.

The only concern is how the structures are depicted in the manuscript (Scheme 1 and 2). The structures should be depicted as figures rather than schemes. In addition, the compound 1d shows up in line 184, before Scheme 2 is referred in the main text. It would be better if the structures in scheme 1 and 2 are combined and assigned as Figure 4.

Overall, the manuscript is the one with high impact and scientific soundness.

Author Response

From our point of view it is reasonable to leave the structures in Scheme 1 and Scheme 2. It is correct that structure 1d shows up before Scheme2 is referred. Since Scheme 2 is already on the next page it is close enough.